# Synthesis, Crystal Structures, Lipophilic Properties and Antimicrobial Activity of 5-Pyridylmethylidene-3-rhodanine-carboxyalkyl Acids Derivatives

**DOI:** 10.3390/molecules27133975

**Published:** 2022-06-21

**Authors:** Ewa Żesławska, Robert Zakrzewski, Arkadiusz Nowicki, Izabela Korona-Głowniak, Antonín Lyčka, Agnieszka Kania, Krzysztof Kazimierz Zborowski, Piotr Suder, Agnieszka Skórska-Stania, Waldemar Tejchman

**Affiliations:** 1Institute of Biology, Pedagogical University of Krakow, Podchorążych 2, 30-084 Kraków, Poland; ewa.zeslawska@up.krakow.pl (E.Ż.); agnieszka.kania@up.krakow.pl (A.K.); 2Faculty of Chemistry, University of Lodz, Tamka 12, 91-403 Łódź, Poland; robert.zakrzewski@chemia.uni.lodz.pl (R.Z.); a.nowi@o2.pl (A.N.); 3Department of Pharmaceutical Microbiology, Medical University of Lublin, Chodźki 1, 20-093 Lublin, Poland; iza.glowniak@umlub.pl; 4Department of Chemistry, University of Hradec Králové, Rokitanského 62, 500 03 Hradec Králové III, Czech Republic; antonin.lycka@uhk.cz; 5Faculty of Chemistry, Jagiellonian University in Kraków, Gronostajowa 2, 30-387 Kraków, Poland; zborowsk@chemia.uj.edu.pl (K.K.Z.); agnieszka.skorska-stania@uj.edu.pl (A.S.-S.); 6Department of Analytical Chemistry and Biochemistry, Faculty of Materials Science and Ceramics, AGH University of Science and Technology, Mickiewicza 30, 30-059 Kraków, Poland; piotr.suder@agh.edu.pl

**Keywords:** rhodanine, antimicrobial activity, lipophilicity, crystal structure, electron density, N···S interaction

## Abstract

The constant increase in the resistance of pathogenic bacteria to the commonly used drugs so far makes it necessary to search for new substances with antibacterial activity. Taking up this challenge, we obtained a series of rhodanine-3-carboxyalkyl acid derivatives containing 2- or 3- or 4-pyridinyl moiety at the C-5 position. These compounds were tested for their antibacterial and antifungal activities. They showed activity against Gram-positive bacteria while they were inactive against Gram-negative bacteria and yeast. In order to explain the relationship between the activity of the compounds and their structure, for selected derivatives crystal structures were determined using the X-ray diffraction method. Modeling of the isosurface of electron density was also performed. For all tested compounds their lipophilicity was determined by the RP-TLC method and by calculation methods. On the basis of the carried-out research, it was found that the derivatives with 1.5 N···S electrostatics interactions between the nitrogen atom in the pyridine moiety and the sulfur atom in the rhodanine system showed the highest biological activity.

## 1. Introduction

Compounds containing the 2-sulfanylidene-1,3-thiazolidin-4-one core (rhodanine), the synthesis of which was first carried out by Nencki almost 150 years ago [1], have interesting biological properties and therefore are still of interest to researchers in the field of medicinal chemistry [2,3,4]. Such derivatives reveal, *inter alia*, antibacterial [5,6], antifungal [7,8], antiviral [9,10], antitumor [11,12,13,14], anti-inflammatory [15] and anthelmintic [16] activities. Other, non-medical applications of rhodanine derivatives are also known. Among other things, they have been used in dye sensitized solar cells (DSSC) [17].

The special interest of many groups of researchers is focused on rhodanine derivatives containing a carboxyalkyl acid fragment at the N-3 position (Figure 1).

Interesting hybrids of quinazolinone and rhodanine-3-acetic acid were studied by Motta et al. [18] These compounds are potential aldose reductase inhibitors. Amides obtained from rhodanine-3-acetic acid are also inhibitors of aldose reductase and aldehyde reductase [19]. Radwan’s team investigated rhodanine-3-acetic acid as a novel alpha amylase inhibitor [20]. Research into the antibacterial activity of rhodanine-3-acetic acid derivatives was described by Neyadi’s team [21], while Monga’s team investigated the ability of rhodanine-3-acetic acid derivatives to inhibit the activity of pancreatic lipase, which is an enzyme responsible for fat digestion [22].

The synthesis of the simplest compound of this type, rhodanine-3-acetic acid (2-(4-oxo-2-sulfanylidene-1,3-thiazolidin-3-yl) acetic acid) was first carried out by Körner [23]. A very well-examined example of a rhodanine-3-acetic acid derivative is epalrestat, an aldose reductase inhibitor. It is used as a drug to treat diabetic neuropathy in non-insulin dependent diabetes mellitus. Furthermore, investigations into the physicochemical properties and biological activity of epalrestat homologues are also carried out [24].

The ability to inhibit aldose reductase is also demonstrated by rhodanine-3-acetic acid derivatives containing a benzylidene moiety with an additional ester moiety at the C-5 position [25,26].

Notably, our previous studies showed that benzylidene and cinnamylidene derivatives of rhodanine-3-carboxyalkyl acids displayed antibacterial activity [27,28].

Considering the high potential of benzylidene derivatives of rhodanine-3-carboxyalkyl acids with the presence of a bulky, tertiary amine group bounded to the aromatic ring, as promising antibacterial compounds, this study is focused on chemical modifications among different location of nitrogen atom. Thus, we decided to check whether the presence of a heterocyclic nitrogen atom in the aromatic fragment would significantly improve the antibacterial activity. Furthermore, we have examined the influence of the position of the heterocyclic nitrogen atom in the aromatic system and the influence of the linker length between the carboxyl group and the nitrogen atom at the position 3 on the antibacterial activity.

In the presented work, we describe the performed synthesis of a series of derivatives of rhodanine-3-butyric acid, rhodanine-3-valeric acid, rhodanine-3-caproic acid and rhodanine-3-undecylic acid containing one of the following groups at the C-5 position: pyridin-2-ylmethylidene, pyridin-3-ylmethylidene or pyridin-4-ylmethylidene group. (Figure 1). We have tested their effect on ten strains of Gram-positive bacteria, five strains of Gram-negative bacteria, and five strains of yeast.

This paper presents synthesis of 12 novel compounds, structural analysis, including X-ray structure analysis for three derivatives, their lipophilic properties and antimicrobial activity.

## 2. Results and Discussion

### 2.1. Chemistry

The synthesis of rhodanine-3-butyric acid, rhodanine-3-valeric acid, rhodanine-3-caproic acid and rhodanine-3-undecylic acid was carried out according to the procedure previously developed for the synthesis of rhodanine-3-acetic acid and rhodanine-3-propionic acid [29].

The Knoevenagel type condensation of the obtained rhodanine-3-carboxyalkyl acids with 2, 3 and 4-pyridinecarboxaldehyde, shown in Figure 1, was carried out according to the procedure previously described during the synthesis of epalrestat homologues [30].

### 2.2. Lipophilicity

Thin-layer chromatography was used to determine the lipophilicity of the tested compounds. The R_f_ parameter was calculated using the following equation:(1)$$R_{f}=\frac{a}{b}$$
where a is the migration distance of the spot (centre of the spot) from the starting line of the plate and b is the total migration distance of the front of the solvent from the starting line of the plate.

Values of R_f_ were used to calculate coefficients R_M_ according to Bate-Smith and Westall equations [31]:(2)$$R_{M}=\log\left(\frac{1}{R_{f}}-1\right)$$

Usually, the R_M_ values determined by thin layer chromatography in the reversed phase system are linearly dependent on the percentage content of organic modifier in the mobile phase, which is presented by the following Soczewiński-Wachtmeister equation [32]:(3)$$R_{M}=R_{M}^{0}+A\varphi$$
where φ is the volumetric fraction of an organic solvent (in these tests methanol, acetonitrile, acetone, propan-2-ol, 1,4-dioxane) in the mobile phase, A is the slope coefficient and $R_{M}^{0}$ is the extrapolated value when pure water is used as the mobile phase. The results obtained, i.e., $R_{M}^{0}$, SD $R_{M}^{0}$, A, SD A and R^2^ (coefficient of determination of the simple equation), for all organic modifiers are shown in Appendix A in Appendix A.

Appendix A shows the results of the calculated Log *p* values from five Internet databases, i.e., ALOGPs, AClogP, XlogP2, XLOGP3 and LogP_ACD_. Each of them automatically generates a value based on the structure of a given chemical.

Physicochemical properties of the analysed compounds such as miLogP, TPSA, MW, NOHBA, NOHBD, V, NORB and NV were obtained from Molinspiration Internet database. The program calculates these parameters based on the structure of the analysed compounds. The results are shown in Appendix A.

In order to assess the bioactivity of the analysed compounds, values of such parameters as GPCR ligand, Ion channel modulator, Kinase inhibitor, Nuclear ligand receptor, Protease inhibitor and Enzyme inhibitor were determined on the basis of the Internet Molinspiration database. The results are shown in Appendix A.

#### 2.2.1. Lipophilicity Parameters

High correlation coefficients (R^2^ > 0.8240) were obtained in simple equations, dependence of retention factor (R_M_) on the percentage content of organic modifier in mobile phase in organic-water modifier system. In 77.2% of cases R^2^ values are higher than 0.9000. For mobile phase organic modifiers such as methanol and 1,4-dioxane the correlation coefficients for each analysed compound are higher than or equal to 0.9733.

As the amount of organic modifier in the mobile phase increased regularly, a regular decrease in R_M_ was observed, therefore A < 0 values for all compounds in all organic-water modifier systems. The highest A-value was obtained for the organic modifier acetonitrile (−0.0156) whereas the lowest for 1,4-dioxane (−0.0727).

The $R_{M}^{0}$ values obtained by reversed-phase thin layer chromatography allow for a very good estimation of the lipophilicity of chemical compounds. A significant effect of applied organic modifier (acetonitrile, methanol, acetone, propan-2-ol, 1,4-dioxane) in mobile phase on the obtained $R_{M}^{0}$ values was observed. Generally, the highest $R_{M}^{0}$ values were obtained for methanol—water system whereas the lowest ones for propan-2-ol—water system and they are in the range 6.4025–2.5845 and 3.1220–1.2167, respectively.

The elution strength in thin layer chromatography in RP-TLC system depends on the solvent polarization—if the lower the value of the polarization index P’ then the higher the elution strength is [33].

For the analysed solvents, propan-2-ol has the smallest polarity index (3,9), which makes it characterized by the highest elution strength. This is confirmed by the studies, because $R_{M}^{0}$ values are the smallest (Appendix A). Methanol and propan-2-ol are in the second group, 1,4-dioxane and acetone in the fourth one and acetonitrile in the sixth group of solvents based on the Snyder triangle selectivity [34].

In the second group, in the Snyder triangle, stronger affinity of compounds to propan-2-ol than to methanol is observed, whereas in the fourth group, stronger affinity to acetone than to 1,4-dioxane is observed. This depends on the proton-donor and proton-acceptor properties of solvents [35].

Comparing the $R_{M}^{0}$ values of the compounds with the different length of the alkyl chain in the molecule (having a nitrogen atom in the same position), the $R_{M}^{0}$ values increase in most cases as the carbon chain length of the molecule increases. For compounds containing a 10-carbon chain the values are much higher than the others (Appendix A). If the longer the alkyl chain, than the more non-polar the molecule is, and lipophilicity is measured by the tendency of chemical compounds molecules to dissolve in fats, oils, and non-polar solvents. The same relationships can be observed by comparing the Log *p* values obtained in-silico (Appendix A).

The distribution of substituents in the molecule may have a large impact on its physicochemical properties. The analysed compounds differed in the substitution of nitrogen atom in the benzene ring. From the obtained experimental data, it can be concluded that the position of the nitrogen atom (in compounds with the same alkyl chain length) in the ring affects (more or less) the $R_{M}^{0}$ values, but these changes are not always the same–sometimes the values for the compounds /**4a–d**/ *meta* are greater than for the compounds /**5a–d**/ *para*, and sometimes the opposite. Usually the compounds /**3a–d**/ with the nitrogen atom in the *orto* position have the highest R_M_ values in the respective series. The results obtained theoretically confirm the relationship that compounds with the same length of alkyl chain differing in the position of the nitrogen atom in the aromatic ring have the highest Log *p* values when the nitrogen atom is in the *orto* position. It is a group of derivatives /**3a–d**/. In the case of compounds/**4a–d**/ with the nitrogen atom in the *-meta* position and compounds /**5a–d**/ with the nitrogen atom in the *para* position the values are close to each other. The Log *p* values of compounds with the same length of alkyl chain vary depending on the position of the atom in the ring and increase according to the order: *meta* < *para* << *orto* (Appendix A). Data obtained from the Molinspiration program looks as follows: *para* < *meta* < *orto* (Appendix A). In the case of LogP_ACD_ results, compounds can be ranked according to increasing lipophilicity as follows: *para* < *orto* < *meta*.

The compound /**3d**/ has the highest $R_{M}^{0}$ values when the mobile phase modifier is acetone, propan-2-ol and 1,4-dioxane and they equal to 4.8853, 3.1220 and 5.5999, respectively. When acetonitrile is used, the highest $R_{M}^{0}$ value is for /**5d**/ (3.4900) and when methanol is used for /**4d**/ (6.4025). All these compounds have a 10-carbon chain in their structure. The compound /**5a**/ has the lowest $R_{M}^{0}$ when the modifier used is methanol, acetone and propan-2-ol and equals to 2.5845, 2.0082 and 1.2167, respectively. When 1,4-dioxane is used, the lowest $R_{M}^{0}$ is for /**4a**/ being 2.7832. All these compounds have a 3-carbon alkyl chain. After the application of acetonitrile, the lowest value has a compound /**4c**/ (1.3658), which is a compound with 5-carbon chain. Values < 2000 also have compounds /**3a**/ and /**5a**/, potentially the least polar. Very similar relationships are visible with in-silico results. The highest Log *p* values are calculated for compounds /**3d**/ (in four cases) and /**4d**/ (in one case) depending on the software used. The smallest Log *p* values have compounds /**4a**/ and /**5a**/ in all cases. This shows that the most lipophilic compounds (the highest $R_{M}^{0}$ or Log *p* values) are /**4d**/, /**3d**/ and /**5d**/, while the most hydrophilic compounds (the lowest $R_{M}^{0}$ and Log *p* values) are /**4a**/ and /**5a**/.

#### 2.2.2. Lipinski Rule of Five and Veber Rules

Christopher Lipinski and his colleagues [36] formulated such 4 criteria that determine a good bioavailability of a chemical compound in the body.

MW (molecular weight) < 500;Log *p* < 5;NOHBD (numbers of hydrogen bond donors) < 5;NOHBA (numbers of hydrogen bond acceptors) < 10.

On this basis, it is possible to assess the probability that a given chemical compound with certain physicochemical properties may be pharmacologically or biologically active, thus becoming a potential oral drug for humans. These rules are not rigid, but allow for a preliminary assessment of whether or not a given compound is bioactive.

The results clearly show (Appendix A) that all analysed derivatives of 3-carboxyalkyl-rhodanine derivates meet the above conditions. The molecular weights are in the range 308.38–406.57, the miLog *p* values are in the range 0.39–4.05, NOHBA and NOHBD for each compound are 5 and 1, respectively. This suggests initially that all compounds should be potentially biologically active and have an effect on the organism when administered orally.

Lipinski’s rules are extended by rules developed by Veber and colleagues [37]. For a compound to show good bioavailability, it must meet only two conditions:number of rotable bonds (NORB) < 14;total polar surface area (TPSA) < 140 Å^2^.

The smaller total polar surface area and the number of rotational bonds is, the higher the penetration rate is. The permeation rate threshold is a prerequisite for good bioavailability in the oral cavity. All tested compounds meet these principles. Total polar surface area, estimated as the sum of the surface area of all polar atoms, mainly oxygen and nitrogen, including the hydrogen atoms attached to them, is 72.20 for each derivative, because the main difference in the structure of the analysed molecules is firstly the length of the non-polar hydrocarbon chain and secondly the position of the nitrogen atom in the aromatic ring. None of these changes in the structure of the molecule affects the value of total polar surface area. The number of rotation bonds is less than 12, thus these molecules have a fairly limited rotation capacity, and therefore potentially good bioavailability. All conditions laid down by the Lipinski and Veber rules are met.

#### 2.2.3. Bioactivity

Estimating the bioactivity of new substances poses a huge challenge to researchers in different scientific fields. Interdisciplinary teams, including chemists, biologists, doctors, pharmacists, etc., have to assess, on the basis of many studies, whether or not a substance can be used as a new potential drug. The parameters described above, such as, e.g., molecular weight, NORBA, NORBD, log *p* is a valuable but very simple indicator of the assessment. They can only be used for preliminary analysis. On their basis it is not possible to determine the substance behaviour in the living organism, its transport, absorption, distribution, protein affinity, bioavailability, toxicity, metabolic stability, etc. However, on the basis of all these parameters, which depend primarily on the structure of the molecule and its physicochemical properties, several additional, valuable parameters can be determined by computer.

The values of such biochemical parameters as GPCR ligand, ion channel modulator, kinase inhibitor nuclear receptor ligand, protease inhibitor and enzyme inhibitor have been determined from the Molinspiration Internet Database [38]. The results are presented in Appendix A. A molecule shows biological activity if the values of these parameters are greater than 0, the values in the range (−5.00)–(0.00) indicate moderate activity, whereas the values < (−5.00) indicate that the molecule is most likely not active [39,40]. The obtained values are in the range (−0.07)–(−1.71), which makes it possible to conclude that all analysed compounds are moderately active. A longer hydrocarbon chain influences the fact that the values are closer to 0, therefore the introduction of additional, non-polar hydrocarbon groups increases the absorption of molecules by the body. The longer the non-polar chain, the more active the molecule is. The position of the nitrogen atom in the benzene ring, in case of molecules with such a structure, as shown in the RP-TLC study, has no major impact on the bioactivity of the molecules. These changes are small, irregular and are incomparably less important than the presence of non-polar substituents.

### 2.3. Crystal and Molecular Structures of /** *4a* **/, /** *5a* **/ and /** *3c* **/

The molecular geometry in the crystals of /**4a**/, /**5a**/ and /**3c**/ with the atom numbering scheme is shown in Figure 2.

The investigated compounds can form two isomers (E or Z) due to the presence of the double bond C5=C6. In the crystal structures the Z isomer is observed for all presented compounds. The fragment of molecules containing the rhodanine and pyridine rings is almost planar. The angle between the planes of these rings are 7.8(1)°, 4.7(1)° and 2.0(2)° for /**4a**/, /**5a**/ and /**3c**/, respectively. Compounds /**4a**/ and /**5a**/ contain propylene linker between carboxyl group and rhodanine ring, while /**3c**/ pentylene linker. Hydrocarbon chains adopt bent conformations with torsion angles N3-C12-C13-C14 and C12-C13-C14-C15 being 169.3(2)°, 62.1(2)° and −72.5(2)°, −76.2(2)°, for /**4a**/ and /**5a**/, respectively. The bent conformation of the linker was also observed in the crystal structure of rhodanine-3-butyric acid [29]. In case of /**3c**/ the hydrocarbon linker is mostly extended with torsion angles close to 180°, except one torsion angle C14-C15-C16-C17 with value of −77.8(3)°.

In the crystal structures of other rhodanine-3-alkylenecarboxylic acid derivatives determined previously, we often observed the characteristic homosynthons, built by two molecules connected into a dimer by strong hydrogen bonds O-H∙∙∙O formed by the carboxyl groups [28,29,30,41] or heterosynthons built by the carboxyl group and a solvent molecule [28,30]. In the presented crystal structures of /**4a**/ and /**5a**/, we have obtained homosynthons built by two molecules connected into a dimer by strong hydrogen bonds O-H∙∙∙N, additionally stabilized by C-H∙∙∙O contacts (Figure 3a,b). For /**3c**/, which also possesses pyridine ring, the dimer is formed by interactions of two carboxyl groups (Figure 3c). The nitrogen atom in position 2 of pyridine ring is not engaged in any intermolecular contacts. The parameters of the intermolecular interactions are listed in Table 1. The neighboring dimers are connected to each other by weak C–H···O and C–H···S intermolecular interactions.

The addition of a pyridine ring to the rhodanine system results in the possibility of sulfur-nitrogen interactions in the studied compounds. Such interactions were described as important in the design of new drugs [42]. Among the compounds examined in this study, such interactions are only possible for compounds /**3a**–**d**/. Some parameters describing the interaction between the sulfur and nitrogen atom are gathered in Table 2. The existence of N∙∙∙S interaction is legitimized by the occurrence of a bond critical point (BCP) between these two atoms. The BCP is the minimum on the electron density path connecting two nuclei. Its presence indicates a strong interaction (chemical bond) joining two atoms in the molecular system [43]. On the other hand, no BCPs are observed between a sulfur and a hydrogen atom in compounds /**4a**/ or /**5a**/. The aforementioned interaction has quite a long distance (around 2.85 Å) and its force constant value is more or less 0.15. The interaction seems to be electrostatic, due to quite high atomic charges difference between nitrogen (the nitrogen atomic charge is about −1.15) and sulfur (the charge is a bit above 0.30) atoms.

Such a picture of the nitrogen-sulfur interaction is supported by the Molecular Electrostatic images presented in Figure 4. The MEP maps for the compounds of the /**3**/ series are compared with the data calculated for the /**4a**/ and /**5a**/ compounds. The maps presented indicate a concentration of electronic potential of different signs around nitrogen and sulfur atom for the compounds of the series /**3**/. Such a concentration is not observed in a similar area of the studied system for other compounds.

### 2.4. Antimicrobial Activity

The obtained compounds revealed no effect neither on the tested Gram-negative bacteria nor the yeast (Appendix A). They showed antibacterial activity directed only against selected strains of Gram-positive bacteria (Table 3). They were active against bacteria belonging to the strains *S**taphylococcus*, *Bacillus* and *Micrococcus*. On the other hand, against bacteria belonging to the genus *Streptococcus* tested compounds showed no or mild bioactivity.

There are few reports in the scientific literature describing the biological activity of rhodanine derivatives containing a pyridine fragment in the C-5 position. Esswein et al. described the use of (4-oxo-5-pyridin-2-ylmethylidene-2-thioxo-thiazolidin-3-yl) acetic acid for the prevention of metabolic bone disorders [44].

Jampilek et al. studied the influence {(5Z)-[4-oxo-5-(pyridin-2-ylmethylidene)-2-thioxo-1,3-thiazolidin-3-yl]}acetic acid, {(5Z)-[4-oxo-5-(pyridin-3-ylmethylidene)-2-thioxo-1,3-thiazolidin-3-yl]}acetic acid and {(5Z)-[4-oxo-5-(pyridin-4-ylmethylidene)-2-thioxo-1,3-thiazolidin-3-yl]}acetic acid on the development of fungi. They found that only the first of the tested compounds significantly inhibited development of *Candida tropicalis* 156, *Candida krusei* E28, *Candida glabrata* 20/I and *Trichosporon asahii* 1199 [45]. Their observations indicate a clear relationship between the antifungal activity and the position of the nitrogen atom in the pyrilidene substituent.

An interesting comparison of the antibacterial activity of rhodanine derivatives and thiazolidine-2,4-dione was performed by Abell et al. His team, examining the effect of the obtained compounds on the *Staphylococcus aureus, Staphylococcus epidermidis* and *Bacillus subtilis* strains, concluded that rhodanine derivatives showed generally better antibacterial activity than their oxygen analogues. They also noticed that rhodanine and thiazolidine-2,4-dione derivatives containing a π-deficient 3-pyridyl group at the C-5 position did not show significant biological activity [46].

Kratky et al. examined the action of 2-[4-oxo-5-(pyridin-2-ylmethylidene)-2-thioxothiazolidin-3-yl] acetic acid and the appropriate amide on microorganisms from the family *Mycobacterium* (*Mycobacterium tuberculosis*, *Mycobacterium avium*, *Mycobacterium kansasii*); however, the obtained results were not satisfactory [47].

In this study, the highest antimicrobial activity against Gram-positive bacteria was shown by compounds /**3a**–**d**/ containing the pyridin-2-ylmethylidene group at the C-5 position (Appendix A). The measured MIC values were 7.8 to 125 µg/mL. Only for this group of tested compounds, we observed a clear change in antimicrobial activity depending on the number of methylidene groups in the linker between the carboxyl group and the nitrogen atom in the N-3 position of the rhodanine ring. The lowest activity was shown by derivatives with n equaling 3 and the highest activity by derivatives with the linker containing ten carbon atoms. The observed changes in activity were consistent with the calculated changes in the LogP_ACD_ parameter. For compound /**3a**/ (*n* = 3), the value of the LogP_ACD_ parameter equals 0.53 and for compound /**3d**/ (*n* = 10), the value of the LogP_ACD_ parameter equals 3.72 (Appendix A). High lipophilicity value is one of the factors facilitating the penetration of chemical compounds into cells [48]. The significant differences in lipophilicity between compounds from the group of /**3a–d**/ allow us to assume that this parameter is responsible for changes in biological activity between them. For the 5-pyridylene rhodanine derivatives, studied previously by Zacchino et al., containing only a hydrogen atom in the N-3 position, no relationship between biological activity and the value of the Log *p* parameter was observed [49]. We suppose that the presence of the carboxyl group in the N-3 position of the rhodanine system is the factor contributing to the increased antimicrobial activity.

The /**4a**–**d**/ derivatives with the pyridin-3-ylmethylidene group in the C-5 position showed the lowest activity inhibiting the growth of Gram-positive bacteria. They were active in the concentration range from 15.6 to 500 µg/mL (Appendix A). This is consistent with the observations of Abell’s team, who investigated the antimicrobial activity of, among others, 5-(pyridin-3-ylmethylidene)-2-sulfanylidene-1,3-thiazolidin-4-one [46]. In this group of compounds, we also observed the impact of the length of the linker in the N-3 position on the activity against Gram-positive bacteria (Appendix A).

The compounds /**5a**–**d**/ containing the pyridin-4-ylmethylidene group in the C-5 position showed a higher inhibitory activity against Gram-positive bacteria than the compounds /**4a**–**d**/ but significantly lower than the compounds /**3a**–**d**/. It was in the range of 7.8 to 500 µg/mL (Appendix A). In this group of compounds, the value of the LogP_ACD_ parameter increased from 0.40 for /**5a**/ to 3.59 for /**5d**/ (Appendix A), however, no correlation was observed between the bacteriostatic activity and the length of the linker in the N-3 position. This effect suggests that their poor antimicrobial activity is not due to the possibility of penetration into the cells of pathogenic microorganisms, but is a result of another mechanism.

The earlier studies have shown that during the synthesis of rhodanine derivatives containing the pyridylmethylidene group in the C-5 position, the Z-isomers with higher stability than the E-isomers are formed. Derivatives of rhodanine analogues such as thiazolidine-2,4-dione, 2-thiohydantoin and hydantoin are synthesized in the same way [50]. The determined crystal structures confirmed the same type of isomerism around the C5=C6 bond (Figure 2). As a consequence of this type of isomerism, in the group of compounds /**3a**–**d**/ the nitrogen atom, which is a part of the pyridine ring, is closer to the S-1 atom compared to the compounds /**4a**–**d**/ and /**5a**–**d**/. This makes an opportunity of the influence on the active centers of enzymes or metal ions through the nitrogen atom and the sulfur atom, as in case of N-(1,3-thiazol-2-yl) pyridin-2-amine derivatives (Figure 5), kinase insert domain receptor inhibitors (KDR). Meanwell et al. [42] analyzed the influence of the position of the nitrogen atom in relation to the sulfur atom on their biological activity. The 2-pyridinyl analogs were 34-fold more active than the 3-pyridinyl analogs. Moreover, the 1,5-N∙∙∙S interaction stabilized the structure shown in Figure 5, which was geometrically similar to the structures determined for the rhodanine derivatives we studied, presented earlier (Figure 2c).

A similar arrangement of atoms occurs in case of 5-(2-pyridylmethylidene)hydantoin, which is a nitrogen analogue of 5-(2-pyridylmethylidene)rhodanine and forms stable complexes with transition metal ions [51], as well as in case of sulfur base analogs of Schiff bases, which are good chelate ligands [52,53].

## 3. Experimental Sections

### 3.1. Materials and Methods

The reagents and solvents necessary for the synthesis of rhodanine derivatives were obtained from Sigma-Aldrich and were used without further purification. Melting points (uncorrected) were measured using a Boetius apparatus. The infrared spectra were recorded using a Jasco FTIR-670 Plus spectrophotometer (JASCO Corp., Tokyo, Japan) in KBr pellets.

MS spectra were obtained using an AmaZon SL mass spectrometer (Bruker Daltonics, Bremen, Germany).

Scanning was performed in the range of 100–1000 m/z, in the positive ionization mode based on typical ion source settings: heated capillary voltage 4.5 kV, temperature 160 °C, spraying gas flow rate 8.5 l/min, solvent flow rate 3 μL/min. CID fragmentation (MS^n^ where *n* = 2 to 4) was carried out in the ion trap analyser with the aid of helium as a collision gas. The collision energy was changed according to MS^n^ spectra overall quality from ca. 0.6 to 1 eV. The samples were introduced into the mass spectrometer in a slightly acidified (up to 0.1% HCOOH) CH_3_OH:CHCl_3_ (1:1, *v*/*v*) solution.

The ^1^H, proton-decoupled and proton-coupled ^13^C and ^15^N NMR spectra were recorded on a Bruker Avance III HD 400 spectrometer (400.13 MHz for ^1^H, 100.62 MHz for ^13^C, 40.55 MHz for ^15^N) in DMSO-*d*_6_. The ^1^H and ^13^C chemical shifts were referenced to internal TMS (=0.0) and ^15^N chemical shifts to external CH_3_NO_2_ in a coaxial capillary. All 2D experiments (gradient-selected (gs)-COSY, gs-HMQC and gs-HMBC) were performed using manufacturer’s software (TOPSPIN 3.5).

A table of NMR measurement results (Appendix A) is provided in the Appendix A.

### 3.2. General Procedure of Knoevenagel Condensation of Rhodanine 3-Alcanoic Acids with Aldehydes

A mixture of 5.0 mmol of the appropriate rhodanine-3-carboxyalkyl acid, 5.5 mmol of the appropriate aldehyde, 15 mmol of triethylamine and 5 g of 4A molecular sieves were placed in a flask containing 25 cm^3^ of isopropanol and heated to reflux for 5 h. After heating was finished, the solution was filtered and 50 cm^3^ of 2 mol/dm^3^ hydrochloric acid solution was added to the filtrate. The obtained precipitate was filtered off and crystallized from glacial acetic acid.

/**3a**/ *4-[4-oxo-5-(pyridin-2-ylmethylidene)-2-sulfanylidene-1,3-thiazolidin-3-yl]butanoic acid*, m.p. 195–197 °C, yield 22.6%, MS [M+1H]^+1^ 309, IR cm^−1^: 1711.5 C=O, 1165.8 C=S, 1608.3 C=C exo.

/**3b**/ *5-[4-oxo-5-(pyridin-2-ylmethylidene)-2-sulfanylidene-1,3-thiazolidin-3-yl]pentanoic acid*, m.p. 203–205 °C, yield 58.8%, MS [M+1H]^+1^ 323, IR cm^−1^: 1711.5 C=O, 1160.0 C=S, 1609.3 C=C exo.

/**3c**/ *6-[4-oxo-5-(pyridin-2-ylmethylidene)-2-sulfanylidene-1,3-thiazolidin-3-yl]hexanoic acid*, m.p. 162–164 °C, yield 66.2%, MS [M+1H]^+1^ 337, IR cm^−1^: 1709.6 C=O, 1153.2 C=S, 1612.2 C=C exo.

/**3d**/ *11-[4-oxo-5-(pyridin-2-ylmethylidene)-2-sulfanylidene-1,3-thiazolidin-3-yl]undecanoic acid*, m.p. 156–157 °C, yield 67.2%, MS [M+1H]^+1^ 407.1, IR cm^−1^: 1708.6 C=O, 1250.6 C=S, 1609.3 C=C exo.

/**4a**/ *4-[4-oxo-5-(pyridin-3-ylmethylidene)-2-sulfanylidene-1,3-thiazolidin-3-yl]butanoic acid*, m.p. 212–213 °C, yield 56.2%, MS [M+1H]^+1^ 309, IR cm^−1^: 1704.8 C=O, 1185.0 C=S, 1604.5 C=C exo.

/**4b**/ *5-[4-oxo-5-(pyridin-3-ylmethylidene)-2-sulfanylidene-1,3-thiazolidin-3-yl]pentanoic acid*, m.p. 172–175 °C, yield 36.2%, MS [M+1H]^+1^ 323, IR cm^−1^: 1713.4 C=O, 1159.0 C=S, 1607.4 C=C exo.

/**4c**/ *6-[4-oxo-5-(pyridin-3-ylmethylidene)-2-sulfanylidene-1,3-thiazolidin-3-yl]hexanoic acid*, m.p. 178–179 °C, yield 58.2%, MS [M+1H]^+1^ 337, IR cm^−1^: 1702.8 C=O, 1214.9 C=S, 1605.5 C=C exo.

/**4d**/ *11-[4-oxo-5-(pyridin-3-ylmethylidene)-2-sulfanylidene-1,3-thiazolidin-3-yl]undecanoic acid*, m.p. 128–130 °C, yield 92.6%, MS [M+1H]^+1^ 407.1, IR cm^−1^: 1710.6 C=O, 1199.5 C=S, 1604.5 C=C exo.

/**5a**/ *4-[4-oxo-5-(pyridin-4-ylmethylidene)-2-sulfanylidene-1,3-thiazolidin-3-yl]butanoic acid*, m.p. 198–200 °C, yield 32.5%, MS [M+1H]^+1^ 309, IR cm^−1^: 1704.8 C=O, 1169.6 C=S, 1600.3 C=C exo.

/**5b**/ *5-[4-oxo-5-(pyridin-4-ylmethylidene)-2-sulfanylidene-1,3-thiazolidin-3-yl]pentanoic acid*, m.p. 219–220 °C, yield 42.3%, MS [M+1H]^+1^ 323, IR cm^−1^: 1714.4 C=O, 1212.0 C=S, 1603.5 C=C exo.

/**5c**/ *6-[4-oxo-5-(pyridin-4-ylmethylidene)-2-sulfanylidene-1,3-thiazolidin-3-yl]hexanoic acid*, m.p. 158–160 °C, yield 56.4%, MS [M+1H]^+1^ 337, IR cm^−1^: 1705.7 C=O, 1153.2 C=S, 1602.6 C=C exo.

/**5d**/ *11-[4-oxo-5-(pyridin-4-ylmethylidene)-2-sulfanylidene-1,3-thiazolidin-3-yl]undecanoic acid*, m.p. 178–181 °C, yield 79.3%, MS [M+1H]^+1^ 407.1, IR cm^−1^: 1704.8 C=O, 1185.0 C=S, 1601.6 C=C exo.

### 3.3. Antimicrobial Activity In Vitro Assay

All target compounds were screened for antibacterial and antifungal activities by micro-dilution broth method using Mueller-Hinton broth and Mueller-Hinton broth with 5% lysed sheep blood for growth of non-fastidious and fastidious bacteria, respectively, or Mueller-Hinton broth with 2% glucose for growth of fungi. Minimal Inhibitory Concentration (MIC) values of tested compounds were evaluated for the panel of reference microorganism from American Type Culture Collection (ATCC), including: Gram-positive bacteria: *S. aureus* ATCC 25923, *S. aureus* ATCC 6538, *S. aureus* ATCC 43300, *S. epidermidis* ATCC 12228, *M. luteus* ATCC 10240, *B. subtilis* ATCC 6633, *B. cereus* ATCC 10876, *S. pyogenes* ATCC 19615, *S. pneumoniae* ATCC 49619, *S. mutans* ATCC 25175; Gram-negative bacteria: *S. typhimurium* ATCC 14028, *E. coli* ATCC 25922, *P. mirabilis* ATCC 12453, *K. pneumoniae* ATCC 13883, *P. aeruginosa* ATCC 9027 and yeasts: *C. albicans* ATCC 102231, *C. albicans* ATCC 2091, *C. parapsilosis* ATCC 22019, *C. glabrata* ATCC 90030, *C. krusei* ATCC 14243.

The procedure for conducting antimicrobial activity testing was described in detail before [28].

### 3.4. Determination of Lipophilicity Parameters

RP-TLC was used, using five chromatographic systems to determine the lipophilicity parameter ($R_{M}^{0}$) of twelve derivatives and analogues of 3-carboxyalkylrhodanine acid.

TLC experiments were carried out on RP-18 F_254s_ aluminium plates (Merck, Darmstadt, Germany). Distilled water and organic modifier (methanol (J.T. Baker, Arnhem, The Netherlands), acetonitrile (POCH, Gliwice, Poland), acetone (POCH, Gliwice, Poland), 1,4-dioxane (Merck, Darmstadt, Germany), propan-2-ol (POCH, Gliwice, Poland) were used as mobile phases. The content of organic modifier in the mobile phase ranged from 30% to 100% (*v*/*v*) in 10% increments.

The compounds /**3a**–**4d**/ were dissolved in acetonitrile, /**5b**–**5d**/ in acetone and /**5a**/ in methanol. Concentrations of all solutions were 1 mg/mL. 1 µL volumes of solutions were applied on the plates with an interval of 1 cm between the spots using a micropipette (Brand, Wertheim, Germany). The starting line was 0.5 cm from the bottom edge of the plate, while the total length of the chromatogram developed in the horizontal chromatographic chamber (CHROMDES, Lublin, Poland) was 5 cm. After developing, the plates were gently dried in the air and the chromatogram was visualized using a UV lamp (Merck, Darmstadt, Germany) at λ = 254 nm. The arithmetic mean was calculated from three TLC analyses. All experiments were conducted at room temperature. All compounds were protected against light by aluminium foil—both during preparation and chromatographic analysis as well as their further storage.

To determine the lipophilicity parameters (logP) in-silico five Internet databases were used: ALOGPs [54], AClogP [55], XlogP2 [56], XLOGP3 [57] and LogP_ACD_ [58]. Each of them uses different algorithms to calculate octanol/water partition coefficients.

Molinspiration Internet database [38] was used to calculate miLog *p* and physicochemical properties of all analysed compounds, including MW, NOHBA, NOHBD, TPSA, V, NORB and NV. Additionally, G values of protein-coupled receptors ligands, ion channel modulators, kinase inhibitors, ion receptor ligand, enzymes and nuclear receptors were determined, which were used to assess the bioactivity of the studied compounds.

### 3.5. X-ray Analysis

Crystals suitable for an X-ray structure analysis were obtained from isopropyl acetate for /**5a**/, propan-1-ol for /**4a**/ and acetonitryl for /**3c**/ by slow evaporation of the solvent at room temperature.

The intensity data for a single crystal were collected using the Oxford Diffraction SuperNova four circle diffractometer, equipped with the Mo (0.71069 Å) Kα radiation source and graphite monochromator. The phase problems were solved by direct methods using SHELXS program [59] and positions of all non-hydrogen atoms were refined anisotropically using weighted full-matrix least-squares on F^2^. Refinement and further calculations were carried out using SHELXL [60].

The tables containing the crystallographic data for the compounds /**3c**/, /**4a**/ and /**5a**/ are included in the Appendix A.

The hydrogen atoms attached to oxygen atoms were identified on difference Fourier maps, whereas all hydrogen atoms bonded to carbon atoms were included in the structure at idealized positions and were refined using a riding model with U_iso_(H) fixed at 1.2 U_eq_ of C. For molecular graphics MERCURY [61] program was used.

/**4a**/ C_13_H_12_N_2_O_3_S_2_, M_r_ = 308.37, crystal size = 0.31 × 0.11 × 0.03 mm^3^, triclinic, space group P$\bar{1}$, a = 5.3749(3) Å, b = 9.8192(3) Å, c = 13.1564(9) Å, α = 73.323(6)°, β = 78.924(5)°, γ = 82.765(7)°, V = 651.91(7) Å^3^, Z = 2, T = 130(2) K, 8719 reflections collected, 3050 unique reflections (R_int_ = 0.0372), R1 = 0.0408, wR2 = 0.1016 [I > 2σ(I)] and R1 = 0.0536, wR2 = 0.1090 [all data].

/**5a**/ C_13_H_12_N_2_O_3_S_2_, M_r_ = 308.37, crystal size = 0.31 × 0.25 × 0.05 mm^3^, monoclinic, space group P2_1_/c, a = 11.5626(2) Å, b = 7.9659(2) Å, c = 14.5520(3) Å, β = 90.169(2)°, V = 1340.35(5) Å^3^, Z = 4, T = 130(2) K, 56894 reflections collected, 3370 unique reflections (R_int_ = 0.0459), R1 = 0.0297, wR2 = 0.0707 [I > 2σ(I)] and R1 = 0.0376, wR2 = 0.0762 [all data].

/**3c**/ C_15_H_16_N_2_O_3_S_2_, M_r_ = 336.42, crystal size = 0.520 × 0.200 × 0.030 mm^3^, triclinic, space group P$\bar{1}$, a = 5.0766(3) Å, b = 12.3912(8) Å, c = 12.6046(8) Å, α = 98.784(5)°, β = 93.967(5)°, γ = 90.083(5)°, V = 781.66(9) Å^3^, Z = 2, T = 130(2) K, 10738 reflections collected, 3716 unique reflections (R_int_ = 0.0386), R1 = 0.0497, wR2 = 0.1171 [I > 2σ(I)] and R1 = 0.0706, wR2 = 0.1274 [all data].

CCDC 2170590-2170592 contain the supplementary crystallographic data. These data can be obtained free of charge from The Cambridge Crystallographic Data Centre via www.ccdc.cam.ac.uk/data_request/cif (accesed on 4 May 2022)

### 3.6. Computational Details

Calculations of selected compounds were executed at the Density Functional Theory [62] level using the long range corrected CAM-B3LYP hybrid functional [63] and the 6–311++G** basis set [64,65,66]. The Gaussian package was used for this purpose. [67] The sulfur-nitrogen interactions were analysed within the Quantum Theory of Atoms in Molecules (QTAIM) [68] and Molecular Electrostatic Potential (MEP) [69] approaches. The electron density of the systems has been analysed within the Quantum Theory of Atoms in Molecules with the AIMAll program [70]. The location of bond critical points (associated to the presence of bonds) and integration of electron density within atomic basins (to estimate atomic charges) were performed. The molecular electrostatic potential of the stationary points was calculated (with the Gaussian package) and analysed on the 0.001 au electron density isosurface. Wiberg force constants [71] were determined using the Multiwfn program [72].

## 4. Conclusions

New pyridyl derivatives of rhodanine-3-carboxyalkyl acids were synthesized to study the effect of the position of the nitrogen atom in the pyridine ring on the antimicrobial activity. All the obtained derivatives met Lipiński’s and Veber’s rules, what indicated their good bioavailability and the potential biological activity. However, the greatest activity against Gram-positive bacteria was observed only for /**3a**–**d**/ containing 5-(pyridin-2-ylmethylidene) moiety. These active compounds demonstrated the highest lipophilicity, both calculated and experimentally determined one.

X-ray crystallographic studies for selected compounds /**3c**,**4a**,**5a**/ provided an insight into structural properties of these three groups containing nitrogen atom in different positions, confirming Z isomer for investigated derivatives. In case of derivative /**3c**/, 1.5 N···S interaction between the sulfur atom of the rhodanine ring and the nitrogen atom in the pyridine ring (2.82 Å) was observed. Such interactions stabilize the conformation of the molecule and can influence the proteins responsible for the functioning of pathogenic microorganisms.

The possibility of interactions between the sulfur and the nitrogen atoms was confirmed by computational methods. The compounds /**3a**–**d**/ show electrostatic interactions between the negatively charged nitrogen atom and the positively charged sulfur atom. The higher antimicrobial activity of compounds /**3a**–**d**/ compared to the compounds /**4a**–**d**/ and /**5a**–**d**/ results not only from their higher lipophilicity but also from the possibility of interactions between the sulfur and the nitrogen atoms. Such an arrangement of atoms not only stabilizes the molecule, but also makes it possible to interact with proteins simultaneously through the sulfur and the nitrogen atoms.

The obtained results indicate that the compounds /**3a**–**d**/ may be precursors of a new series of derivatives with better antimicrobial activity. Due to the presence of the carboxyl group in the N-3 position of the rhodanine system, it is possible to obtain new derivatives with higher lipophilicity and bioavailability than those studied so far.

## Data Availability

All data generated or analysed during this study are included in this published article. Moreover, the datasets used and/or analysed during the current study are available from the corresponding author on reasonable request.

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
