# Peer review of "Synthesis, Crystal Structures, Lipophilic Properties and Antimicrobial Activity of 5-Pyridylmethylidene-3-rhodanine-carboxyalkyl Acids Derivatives"

_molecules, 2022, doi:10.3390/molecules27133975_

Round 1

Reviewer 1 Report

The article "Synthesis, Crystal Structures, Lipophilic Properties and Anti-2 microbial Activity of 5-Pyridylmethylene-3-Rhodanine-Carbox-3 yalkyl Acids Derivatives" by Tejchman et al. reports new bioactive molecules with verified antimicrobial propeperties. Starting from a liphofilicity study conducted by thin layer chromatography and LogP calculus by some internet databases where the obtained values are based on the structure of the designed chemical, to get new active substances specific against some bacteria strain. Evaluation on these molecules bioavailability considering Lipinski rule of five and Veber rules were done before  the synthesis.  Twelve new rhodanine derived molecules were synthesized and caharacterized by diffraction (SCXRD) and by spectroscopic investigations (FT-IR, H NMR, 13C NMR, 15N NMR). Computation study were made to verify the molecular electronic potential of some selected compounds. These new-synthesis compounds showed antibacterial activity directed only against selected strains of Gram-positive bacteria where no activity is registered for the Gram-negative and the yeast monitored during the experiments. Among the rhodanine derived series of molecules the group 3a-d containing the pyridin-2-ylmethylene group at the C-5 position exhibits the highest antimicrobial activity also enforced by lipophylicity.

I suggest to publish this paper after some minor revisions.

Please check table 3 layout.

Authors could present figure 5 using mercury software selecting and isolating the molecule fragment of interest instead of the molecular scheme as reported.

For an enhanced data presentation I suggest to include some missing spectra and tables in Supporting Information file:

-Add NMR spectra in SI.

-Add IR Spectra in SI.

-Add Crystallographic data tables in SI.

Moreover, conclusion section need to be separated from the previous paragraph and  be written in a clear form and using higher grade english editing.

Author Response

We would like to thank Reviewer 1 for their comments on the manuscript.

The layout of Table 3 has been improved.

Figure 5 is based on the information contained in the publication: Beno, B.R .; Yeung, K.S .; Bartberger, M.D .; Pennington, L.D .; Meanwell, N.A. A survey of the role of noncovalent sulfur interactions in drug design, J. Med. Chem. 2015, 58, 4383–4438. The Chem Sketch program was used to draw the corrected model because Mercury did not allow for the introduction of a general designation of the alkyl group. The dashed green line indicates 1.5 N ··· S interactions.

The assignment of signals in the 1H NMR, 13C NMR, and 15N NMR spectra is presented in the supplement in Table S8. We hope that this form of presenting the results will be satisfactory.

The allocation of the bands in the IR range to the most important functional groups is provided in Section 3.2. together with the reaction yields, melting points, and results of MS analysis.

The tables containing the crystallographic data for the compounds / 3c /, / 4a / and / 5a / have been added to the Supplementary Materials. (Tables S9, S10 and S11).

The text has been corrected by a native speaker.

In a word document, the summary section is separate from the previous paragraph. We do not know why the two paragraphs were merged in PDF format.

Reviewer 2 Report

In this work synthesis, crystal structures, lipophilic properties and antimicrobial activity of 5-pyridylmethylene-3-rhodanine-carboxyalkyl acids derivatives are described. It was shown that the obtained compounds showed activity against Gram-positive bacteria while they were inactive against Gram-negative bacteria. In order to explain the relationship between the activity of compounds and their structure, their crystal structures were determined for a few selected derivatives using the X-ray diffraction method. Taking into account the mentioned below notes I think that this article needs major revision.

Notes:

1. Are the obtained 3a-d, 4 a-d, 5a-d compounds new synthesized? Why NMR spectroscopy was not applied for their characterization? It should be added in the Experimental sections.

2. The yields of new synthesized compounds should be added in the synthesis scheme.

3. Conclusions of this work should be added at the end. Moreover the promising application of new obtained results should be noted.

Author Response

We would like to thank Reviewer 2 for all the comments and suggestions on the manuscript.

ad 1. To ensure the clarity of the text, in point 3.2. we only present the melting points, reaction yields, MS analysis results, and the assignment of IR bands to the most important functional groups. The assignment of the signals in the 1H NMR, 13C NMR, and 15N NMR spectra is presented in the Supplementary Materials in Table S8. We hope this will be satisfactory.

ad 2. It seems to us that including the reaction efficiency in the synthesis scheme will reduce the legibility of the drawing.

ad 3. The Conclusions section has been corrected.

Round 2

Reviewer 2 Report

In this work synthesis, crystal structures, lipophilic properties and antimicrobial activity of 5-pyridylmethylene-3-rhodanine-carboxyalkyl acids derivatives are described. It was shown that the obtained compounds showed activity against Gram-positive bacteria while they were inactive against Gram-negative bacteria. In order to explain the relationship between the activity of compounds and their structure, their crystal structures were determined for a few selected derivatives using the X-ray diffraction method.

Authors have corrected all comments in the paper and quite clearly answered to the questions. I hope these corrections improved the paper and the revised version corresponds to high standards of Molecules. After careful consideration, I think that this paper may be published in this view.